# PG-path: Modeling and personalizing pharmacogenomics-based pathways

**Joo Young Hong**[1,2], **Ju Han Kim**[1,3]*

**1** Division of Biomedical Informatics, Seoul National University College of Medicine, Seoul, Korea,
**2** Cipherome. Inc., Seoul, Korea, **3** Division of Biomedical Informatics, Seoul National University Biomedical Informatics (SNUBI), Seoul National University College of Medicine, Seoul, Korea

* juhan@snu.ac.kr

## Abstract

A pharmacogenomics-based pathway represents a series of reactions that occur between drugs and genes in the human body after drug administration. PG-path is a pharmacogenomics-based pathway that standardizes and visualizes the components (nodes) and actions (edges) involved in pharmacokinetic and pharmacodynamic processes. It provides an intuitive understanding of the drug response in the human body. A pharmacokinetic pathway visualizes the absorption, distribution, metabolism, and excretion (ADME) at the systemic level, and a pharmacodynamic pathway shows the action of the drug in the target cell at the cellular-molecular level. The genes in the pathway are displayed in locations similar to those inside the body. PG-path allows personalized pathways to be created by annotating each gene with the overall impact degree of deleterious variants in the gene. These personalized pathways play a role in assisting tailored individual prescriptions by predicting changes in the drug concentration in the plasma. PG-path also supports counseling for personalized drug therapy by providing visualization and documentation.

**Data Availability Statement:** All relevant data are within the paper and its Supporting Information files.

**Funding:** There is no funding associated with this study. This study was conducted as a Ph.D. thesis work of the first author at Seoul National University

## Introduction

"A picture says more than a thousand words." For many years, pharmacologists have generated pathway diagrams to elucidate the pharmacology of drugs administered to humans. These diagrams have proven to be powerful tools to organize, share and discuss knowledge. [1]

One type of diagram that visually illustrates the flow and actions of a drug in the human body is a pharmacogenomics (PGx)-based pathway, which represents a series of reactions that occur between drugs and genes in the human body after drug administration. [2] Researchers use this type of diagram to describe pharmacokinetics (PK) and pharmacodynamics (PD). A PK pathway represents the absorption, distribution, metabolism, and excretion (ADME) carried out by enzymes, transporters, and/or carriers on a drug at the systemic level. A PD pathway depicts the mechanism of action by which a drug affects its targets, such as components of metabolic pathways or signaling pathways, or the receptors themselves, in the target cell at the cellular-molecular level. [3, 4]

College of Medicine. No funder has a role in study design, data collection and analysis, decision to publish, or preparation of the manuscript.

**Competing interests:** There are no competing interests associated with this study. After having accomplished this study at Seoul National University College of Medicine, the first author is serving as a researcher in Cipherome. Therefore, Cipherome is not related to this study but is currently interested in this study result. These do not alter our adherence to PLOS ONE policies on sharing data and materials.

PGx studies analyze the genes involved in PK and PD and interpret how variations in these genes alter the function of proteins. Based on this interpretation, PGx studies aim to maximize the safety and efficacy of drugs by adjusting the dosage or changing the drugs themselves. [5–7] A PGx-based pathway visualizes the interaction between drug-related genes and a drug in terms of PK and PD and helps us understand how changes in protein function, based on genomic variation, affect the ADME of the drug in the body and the drug action in target cells. [2]

As research on the interaction between drugs and genes has progressed alongside advancements in the Internet and systems biology, [8] many PGx-based pathways have begun to be systematically created. As a result of multidisciplinary studies, many pathway resources have been developed for various goals ranging from the identification of drug-related genes in each model organism to the development of tools for drug discovery. [9–11] Specific drug-related pathway types include human metabolic pathways, metabolite signaling pathways, metabolic disease pathways, and drug-metabolic or drug-action pathways. [12]

Pathway resources that utilize PK and PD in relation to drug-related genes include the Pharmacogenomics Knowledgebase (PharmGKB) [13] and the Small Molecule Pathway Database (SMPDB) [12, 14]. These resources visualize pathways using internally standardized components and formats and are hyperlinked to massive PGx database resources. However, the pathways of many drugs are still not standardized, and separately created pathways are also not standardized with regard to interoperability. Most drug pathways are used to identify the function of genes or search for new genes, but they do not support tailored drug treatments to which PGx concepts are applied in clinical settings.

We propose the new knowledge-based PGx pathway: PG-path. We collected scattered drug pathways and standardized and visualized them to help users intuitively understand the PK and PD of a specific drug. In addition to its use in research, PG-path is designed to be used to assist personalized drug prescriptions by predicting the change in plasma drug concentration using the overall impact degree of deleterious variants in a gene, in the clinical field.

## Materials and methods

### Materials

**DrugBank 5.0.1.** The components of PG-path were generated on reference DrugBank 5.0.1. [15] The components extracted from DrugBank include drug-gene pairs, drug interaction types (enzymes, transporters, carriers, targets), and drug actions (inhibitors, inducers, substrates). This information was derived from extensive primary literature sources by domain-specific experts and skilled biocurators related to DrugBank. [15]

**Sorting Intolerant From Tolerant (SIFT).** Using sequence homology, SIFT predicts whether an amino acid substitution will affect the protein function and consequently alter the phenotype. [16–18]

**The 1000 Genomes Project.** The 1000 Genomes Project is a comprehensive, open-access database of genetic variants from 2,504 individuals from 26 populations across the globe. [19, 20]

**Gene-wise Variant Burden (GVB) score.** Variations in a gene that interacts with a drug can affect the function and activity of the protein encoded by the gene, resulting in an altered phenotype. The altered phenotype affects the drug's ADME, resulting in changed drug concentrations in the plasma, which shows different drug efficacy and toxicity from person to person. The GVB score, which indicates the degree of protein damage, is applied to the pathways, which is one method of the uses of PG-path. This GVB score was developed using the SIFT score. [17] The GVB score, defined as the geometric mean of the SIFT scores for the set of coding variants in a gene, is applied to estimate the overall impact of all deleterious variants in the

gene. [21] This method operates under the hypothesis that variants that potentially change protein function are not assured, but may cause deleterious phenotypes. [22] A lower GVB score represents an increased likelihood of a variant to be more deleterious to the function of the protein encoded by the gene. [21, 22]

## Pathway development methods

Each PK/PD pathway is developed by applying the following method (Fig 1B):

1. Extract the drug-gene pair, drug interaction type, and drug action from DrugBank 5.0.1 for the selected drug (S1 Table)

2. Create a standard frame for the PK/PD pathway and produce a set of symbols (S1A Fig)

3. Draw the nodes and edges of the PK/PD pathway using PathVisio software (S1B Fig)

4. Save the data in the Graphical Pathway Markup Language (GPML) format and store the information in a PK/PD GPML folder

5. Draw the background image for the PK/PD pathway using Illustrator CS6 (S1C Fig) (PK: one background image for all drugs; PD: each drug has its own background image)

6. Save the data in the Portable Network Graphics (PNG) format and store the information in the PK/PD PNG folder

7. Convert the GPML file to the Scalable Vector Graphics (SVG) format using PathVisioRPC (PK/PD pathway)

8. Merge the SVG file with the background image PNG file (PK/PD pathway) (S1D Fig)

9. Link to an SVG + Hypertext Markup Language (HTML) file with gene and drug information windows and a PD description window

10. Generate the complete PK/PD pathway

## Pathway development software

Pathway diagrams were generated in the GPML format using PathVisio 3.2.4, and background images were created in the PNG format using Adobe Illustrator CS6. Each pathway diagram saved in the GPML format was converted to the SVG format using PathVisioRPC, an XML-RPC interface, after code modification.

## Results

### Nodes and edges

PG-path is a human-centered drug pathway, although antibiotic, antifungal, and antiviral drugs are included in PG-paths. Each pathway is produced by dividing the PK pathway, representing the ADME in the human body, and the PD pathway, describing the action of the drug at the target cell. PK pathways have been elucidated at the systemic level based on metabolism and transport, and PD pathways have been characterized at the cellular and molecular- level based on drug actions and responses.

A set of symbols was created to standardize each component of the pathway concerning the role of each component. Drugs are classified as prodrugs and active drugs. *Prodrugs* become active when they are absorbed and metabolized by the human body, and *active drugs* are already active in their present form. These two categories are distinguished by the same shape

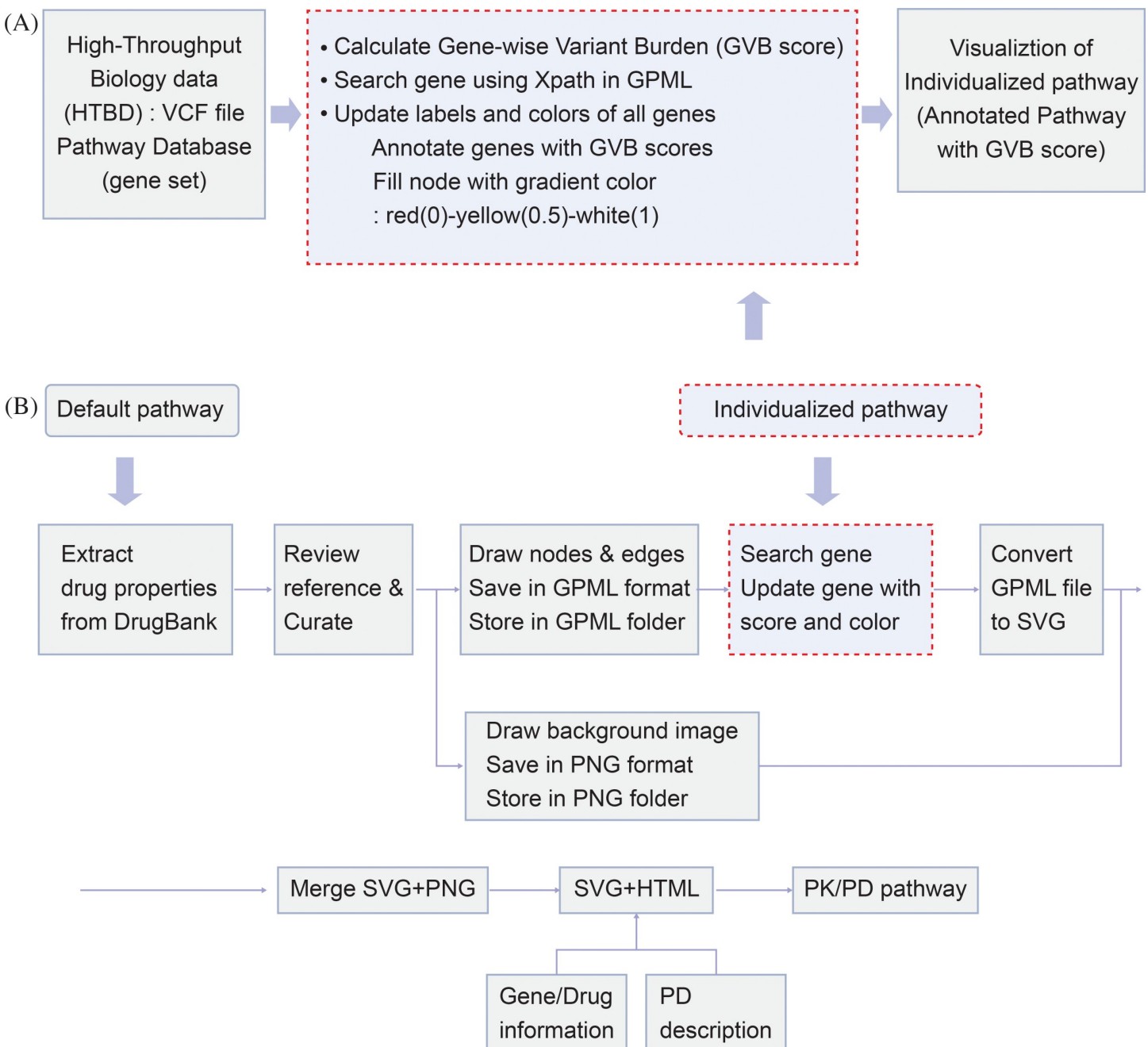

**Fig 1. Pathway analysis and modeling procedure.** (A) Three phases of pathway analysis: (1) input the high-throughput biological data (HTBD) variants from the VCF file; (2) perform the algorithm-based analysis and GVB scoring; (3) visualize the pathway and annotate with the GVB scores. (B) Development procedure for default pathways and personalized pathways. VCF: Variant Call Format; GVB: Gene-wise Variant Burden; GPML: Graphical Pathway Markup Language; SVG: Scalable Vector Graphics; PNG: Portable Network Graphics; HTML: Hypertext Markup Language; PD: Pharmacodynamics.

with different colors. Metabolites produced through enzymatic metabolism of a drug are also divided into inactive metabolites and active metabolites. These two categories are distinguished by the same shape in different colors as a drug. The genes encoding enzymes, transporters, and carriers play different roles; therefore, they differ in shape and color. After defining each node, we standardized the edges representing the actions between the nodes.

The actions consist of metabolism, transportation, binding, and excretion. The edge corresponding to each action is selected using PathVisio's molecular interaction map (MIM) tool. [23] Inhibitors and inducers, action types that refer to the actions of drugs on genes, are also shown as edges. One of the remaining symbols, the 'major' marker, indicates enzymes that play a significant role in metabolism, while the 'active' marker indicates the metabolites that are active when the prodrug is metabolized. (Table 1)

For the symbols of the PD pathway, the target, drug, and active metabolite comprise the nodes, and inhibition, activation, metabolism, binding, conversion, and action comprise the edges. (Table 2)

## Background frame and image

In a PK pathway, a background frame modeled on the human body is utilized so that the proteins can be superimposed on the correct human compartments. A comprehensive range of standard background compartments was selected to precisely express the ADME routes. In the standard PK pathway frame, the background anatomical organs are the eye, nose, mouth, brain, lung, heart, muscle, kidney, liver, adrenal gland, testis, intestines, placenta, and skin, in addition to other compartments that are needed for a few drugs. Drugs are distributed through the arteries and veins and excreted through the bile duct or urinary tract. The drug concentration of absorption differs across routes of administration, so we must be concerned with where and how the drug is administered. The routes of administration include eye drops, inhalation, sublingual and buccal absorption, oral ingestion, intravenous injection, intramuscular injection, and percutaneous absorption. The tissues site at which metabolism or transport occurs depends on the expression levels of enzymes and transporters, which can be found by referring to the Protein Atlas databases. (S1A and S1B Fig) We drew one background image, composed of standardized body compartments, to graphically visualize the ADME of the drug in the PK pathway. (Table 1, S1C and S1D Fig)

The components of each PD pathway include drugs, active metabolites, and genes encoding targets. For visualization of the interactions between the drug and its targets, the PD diagram

**Table 1. Characteristics of a pharmacokinetic pathway.**

| Characteristic | Description |
| --- | --- |
| Drug | active drug, prodrug (activating enzyme) |
| Metabolic element | inactive metabolite, active metabolite |
| Protein | enzyme (e.g., activating enzyme), transporter, carrier |
| Background anatomical organs | eye, nose, mouth, brain, blood-brain-barrier, lung, heart, muscle, skin, kidney, liver, adrenal gland, testis, intestines, and placenta |
| Transport structures | blood vessels (e.g., arteries and veins), bile ducts, and excretory tracts (e.g., urinary tract for urine, gut lumen for feces) |
| Methods of administration | eye drop, inhalation, sublingual and buccal absorption, oral ingestion, intravenous injection, intramuscular injection, and percutaneous absorption |
| Interaction types (from DrugBank) | metabolism, transportation, binding, excretion |
| Action types (from DrugBank) | inhibition, induction, substrate |
| Tissue site for ADME | expression levels of enzymes and transporters through the ProteinAtlas database |

Methods of administration: the location where the drug is administered; Interaction type: the protein role, according to which the reaction between a drug and a gene happens; Action type: the type by which a drug acts to a protein or vice versa; ADME: an abbreviation in pharmacokinetics for Absorption, Distribution, Metabolism, Excretion; DrugBank version, 5.0.1.

**Table 2. Characteristics of a pharmacodynamic pathway.**

| Characteristic | Description |
|---|---|
| Components | active drugs, prodrug, active metabolites, genes (targets) |
| cellular-level component | cell components including the nucleus, endoplasmic reticulum, mitochondria, Golgi apparatus, lysosomes, peroxisomes, vesicles, cell membrane, ribosomes. |
| Action types (from DrugBank) | agonist, antagonist, activator, modulator, competitor, cofactor, ligand, stimulator, antibody, binder, potentiator, neutralizer, inhibitor, inducer, etc. |
| Interaction types (from DrugBank) | the role of the biological pathway of the gene at the molecular level (target) |

Interaction type: the protein role, according to which the reaction between a drug and a gene happens; Action type: the type by which a drug acts to a protein or vice versa; DrugBank version, 5.0.1.

was drawn with PathVisio. The standard biological mechanism inside the target cell was graphically described as a background image at the cellular level. The cellular elements comprise the nucleus, endoplasmic reticulum, mitochondria, Golgi apparatus, lysosomes, peroxisomes, vesicles, cell membrane, ribosomes, etc. In this way, the effect of a drug can be presented intuitively through its mechanism of action on the target. (Table 2)

## Convert and merge

For a pathway diagram created in the GPML format, the attributes of nodes and edges can be changed using XPath. The modified GPML file is converted to the SVG format using PathVisioRPC software. Each drug is retrieved using its Human Metabolome Database (HMDB) identifier, and each gene is retrieved using its HUGO Gene Nomenclature Committee (HGNC) identifier, allowing the identification of nodes using XPath. PathVisioRPC is an XML-RPC interface for PathVisio. This interface enables us to utilize the PathVisio functionality directly in our analytical environment of choice. The modified and transformed pathway diagram and background image PNG file are used to generate a complete PK/PD pathway in SVG + HTML format through the merge function. (Fig 2A, Fig 2C, S1 Fig)

## Information on drugs and genes

Drugs and genes (nodes) in the pathway are linked with detailed information, and the PD pathway is linked with a description to easily explain the mechanism of action of the drug on the target. These knowledge bases are manually curated with various references.

In the merged pathways, genes are hyperlinked to the external HGNC database, and drugs are hyperlinked to the HMDB. The identifier of each node that can be connected to an external database is linked to a window containing detailed information about genes or drugs through code modification. If the drug is clicked, a pop-up window shows clinical information on the drug, such as the drug name, Anatomical Therapeutic Chemical (ATC) class, indications, adverse drug effects, drug interactions, and pharmacokinetic data. (S2A Fig) If a gene is clicked, a pop-up window shows information on the gene and its interaction with the drug, such as the gene symbol, gene name, interaction type, chromosomal location, expression tissue, action type, and gene function description. (S2B and S2C Fig) The PD pathway has a link to an internally generated pathway description window to help users understand the mechanism of action. (S2D Fig)

## Drawing a pathway with PathVisio

It is essential to provide standard terminology and formatting so that prewritten computer programs can transfer the information from the standard format to the local application

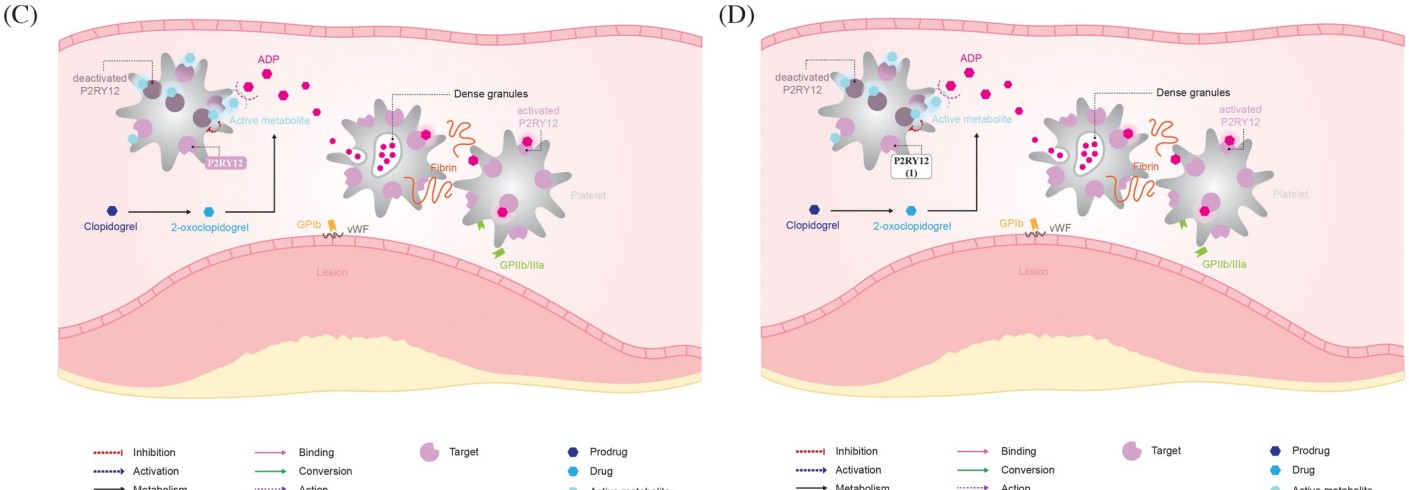

**Fig 2. Pharmacokinetic (PK) and Pharmacodynamic (PD) pathways of clopidogrel.** (A) Default PK pathway; (B) personalized PK pathway with GVB scores of NA12878 sample in the 1000 Genomes Project; (C) default PD pathway; (D) personalized PD pathway with GVB scores of NA12878 sample in the 1000 Genomes Project; GVB: Gene-wise Variant Burden.

format for use, and vice versa. In PGx research, the importance of the format is increasing. The resolution and adoption of the standard formatting will facilitate exchanges between different sources of data. [2]

For the standard format, we built PG-paths using PathVisio, which is a free, open-source, downloadable pathway software. This software has various export formats. The pathway, which is stored in the Biological Pathway Exchange (BioPAX) level3, GPML, and SVG formats, can be modified in various styles or for specific purposes by editing the source code. [1] When pathways are drawn with this software, the attributes (color, size, and labels) of the nodes (genes, drugs, and metabolites) are unified with a standard type. Additionally, the attributes (color and arrowheads) of the edges (metabolism, transportation, binding, excretion) are unified conceptually in accordance with the interaction and action types. Because PathVisio 3.2.4 includes an embedded MIM tool, the interaction and action types can be selected, consistent with a specific purpose. [23] Each pathway is stored in the SVG and GPML formats to allow modification of the source code and interconnection with other databases. Thus, the pathway can be used for computational analysis of genomic variation data or visualization.

## Usage of PG-path: Pathway analysis and visualization with GVB score

Out of the existing gene scoring methods, we chose the GVB scoring method to apply to algorithm-based pathway analysis. Since drug-related genes, unlike disease genes, can be expressed in a variety of phenotypes in healthy individuals, samples were randomly taken from public databases regardless of the specific disease. We applied the GVB scoring algorithm to variants extracted from the Variant Call Format (VCF) file of the NA12878 sample. The NA12878 sample is one of 2504 samples provided in the 1000 Genome Project, which is a public database. [24] We calculated the GVB score of each gene included in the pathway. The calculated GVB scores were then applied to each default pathway to create a personalized pathway. Applying GVB scores to a pathway involves three phases: (1) Inputting the high-throughput biological data (HTBD) (2) Performing the algorithm-based analysis and (3) Visualizing the output data in the pathway. [25] (Fig 1A) We can generate the personalized PK/PD pathway by inserting the GVB scores with gradient colors into a pathway diagram. The Pathway Analysis step is processed between steps 7 and 8 of the default pathway development method. (Fig 1) The personalized PK/PD pathway generated by annotating the GVB score can be used to predict the change in plasma drug concentration according to the score distribution of genes in the human body. (Fig 2B, Fig 2D)

## Discussion

### Prediction of changes in plasma drug concentration by gene placement in pathways

The newly proposed PG-path differs from currently used pathways in that it specifies the location of the genes that interact with the drug when the drug is administered. This is important because in order to explain the change in the concentration of the drug, we must depict as precisely as possible where the gene is located in the human body, and thereby making it easier to understand the general flow of drugs at the system level.

Considering the ADME in general oral drugs, assuming the body is one compartment and the rate of drug elimination in the plasma is constant, the change in plasma concentration of active ingredient from absorption to excretion of the drug varies from person to person, but the pattern is similar. From a pharmacokinetic point of view, when an active drug is administered to the human body, it undergoes ADME and reabsorption. In particular, the amount of oral drug absorbed by the human body is determined by the bioavailability of the absorption

step. Drugs transported to the liver through the portal vein undergo the first-pass effect and are primarily converted to inactive metabolites and excreted. The remaining amount of drug is distributed in highly perfused tissues, such as the brain, heart, liver, kidney, and red blood cells, or slowly infused tissues, such as fat, skin, and muscle. After reacting at the target site, the drug is metabolized in the liver and excreted through bile or urine. During excretion, a small amount of active drug returns to the plasma through reabsorption of the gastrointestinal or proximal tubules. Not all drugs have the same pharmacokinetic pattern. The change in the plasma concentration of the active ingredient depends on whether it is an active drug or a pro-drug. In general, active drugs that are efficacious at the target site are excreted after metabolism in the liver, while prodrugs undergo the first-pass effect, i.e., they are metabolized and acti-vated to show efficacy at the target site. Many factors affect drug response from a pharmacody-namic perspective, but we only consider the concentration and retention time of the active ingredient in the target cell in connection with pharmacokinetics. From a pharmacogenomics perspective, genetic factors in enzymes, transporters, carriers, and targets, in addition to the usual drug reaction and elimination processes, have an essential effect on the plasma concen-tration of active ingredients. The impaired protein's location on the pathway makes it easy to understand where the problem is in the ADME process and to predict changes in plasma drug concentration. [26] (Fig 3)

For the NA12878 sample, Clopidogrel is a prodrug, and considering the GVB score, it can be seen that the score of CYP2C9 among liver enzymes is quite low. At the point where the first pass effect occurs, the GVB score of the enzyme protein that converts the prodrug into an active metabolite is significantly lower, decreasing the formation of the active ingredient which causes the drug response. In other words, although CYP2C9 is not one of the major enzymes for clopidogrel metabolism, it is assumed that the metabolic capacity is low because it has dele-terious variations. It can be predicted that active metabolites will be produced at below-average levels in the liver and that the drug will not reach the maximum therapeutic area. (Fig 2)

Therefore, we can predict the change of pharmacokinetics and pharmacodynamics because the new PG-path specifies the location of the genes that interact with the drug when the drug is administered, and we can expand the understanding of the drug effects and toxicity, together with genetic variations.

**PG-path: One drug centered pathway.** The newly proposed pathway allows the depiction of only one drug at a time, considering the drug metabolism and drug response, which indi-cates that the PG-path does not represent the whole metabolic pathway or the entire signaling pathway. We must concentrate on the range of analysis at the single drug scale because person-alized therapy focuses on administered drugs only. [25]

For example, the new pathway can only focus on the genes associated with a single drug, making it easy to see which genes the drug inhibits or induces in the pathway. In addition, observing two drugs taken together can focus on the two drugs separately, and the drug-drug interaction (DDI) between the two drugs can be seen from a genetic perspective, which allows for drug-gene centric analysis. If Drug A inhibits gene E and Drug B becomes the substrate of gene E, the metabolism of Drug B, which is the substrate, slows down when Drug A and Drug B are taken together. In the case of multidrug, the PG-paths can aid in predicting whether the drug effects will increase or decrease. [27] (Fig 4)

## Pathway analysis: GVB scoring method using variations obtained from DNA sequencing

In terms of pathway analysis, unlike previous analyses using gene expression data obtained from microarray or RNA sequencing [28], PG-path uses the GVB scoring method, which

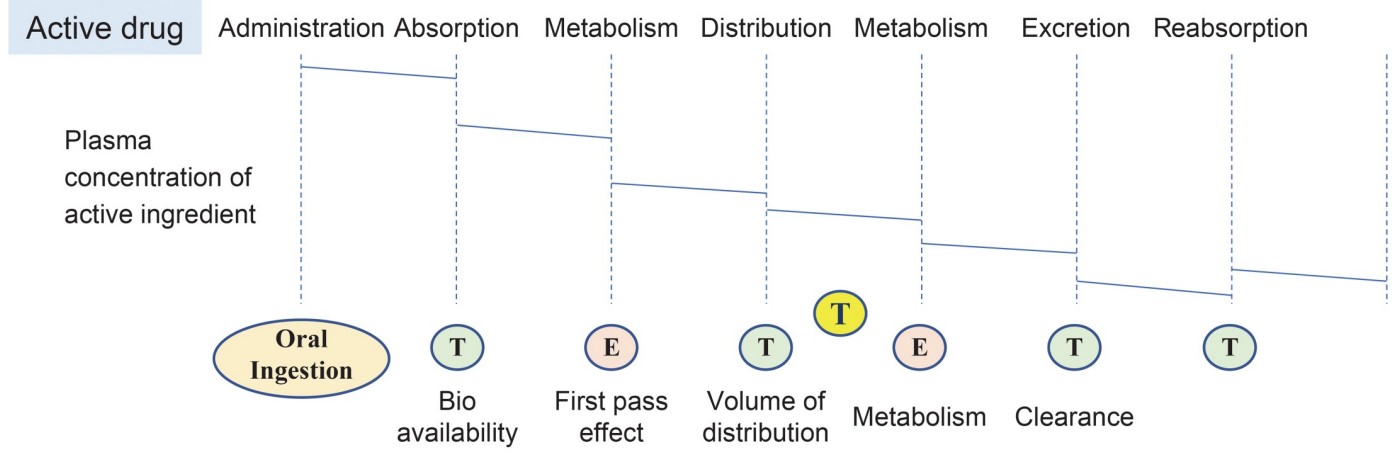

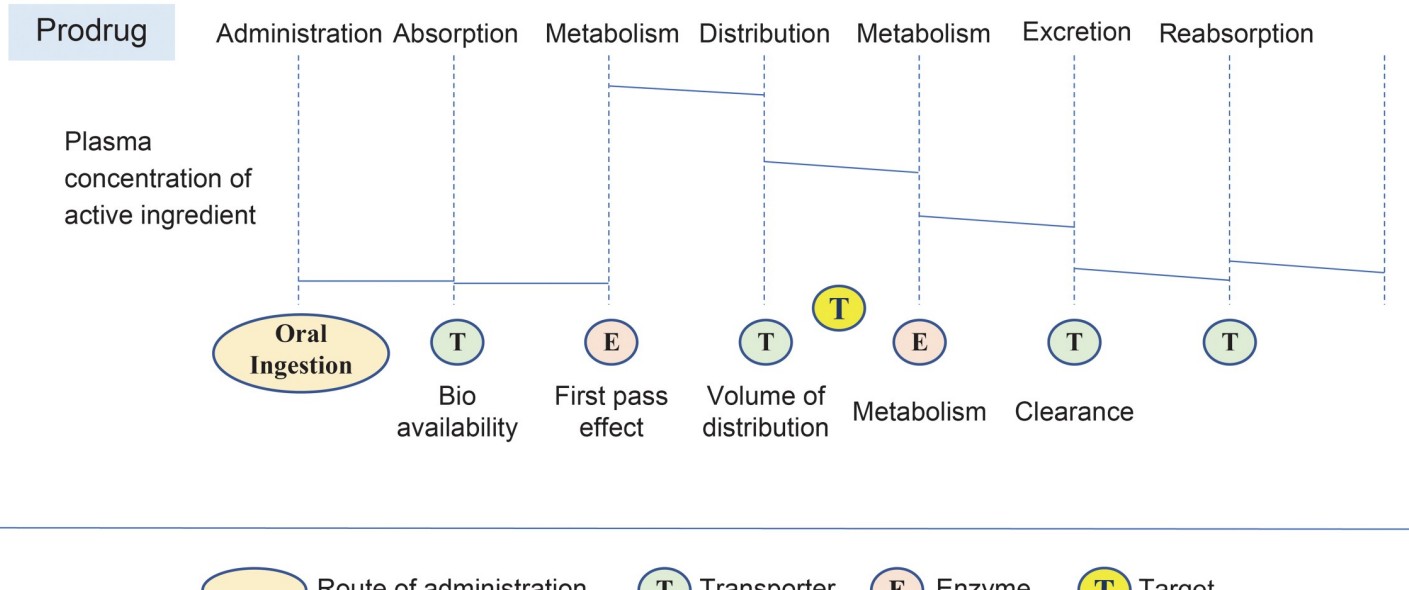

**Fig 3. The change in the plasma concentration of active ingredient depends on drug type.** Assuming the body is one compartment and the rate of drug elimination in the plasma is constant, when a drug is administered to the human body, it undergoes absorption, metabolism, distribution, excretion, and resorption. In this stage, the plasma concentration of an active ingredient is decreased step by step. However, in the case of a prodrug, after absorption and metabolism, the plasma concentration of an active ingredient appears, and the next steps are the same as those of the active drug.

aggregates deleterious variant scores from the DNA sequencing results obtained through next-generation sequencing (NGS). [29] Here, this algorithm-based method applies the altered degrees of the structure and function of the protein encoded by a gene to the gene set interacting with the drug. We then predict the change in drug concentration by considering the location of the genes. Pathway analysis can convert massive variations in genome data into meaningful interpretations of plasma drug concentration changes.

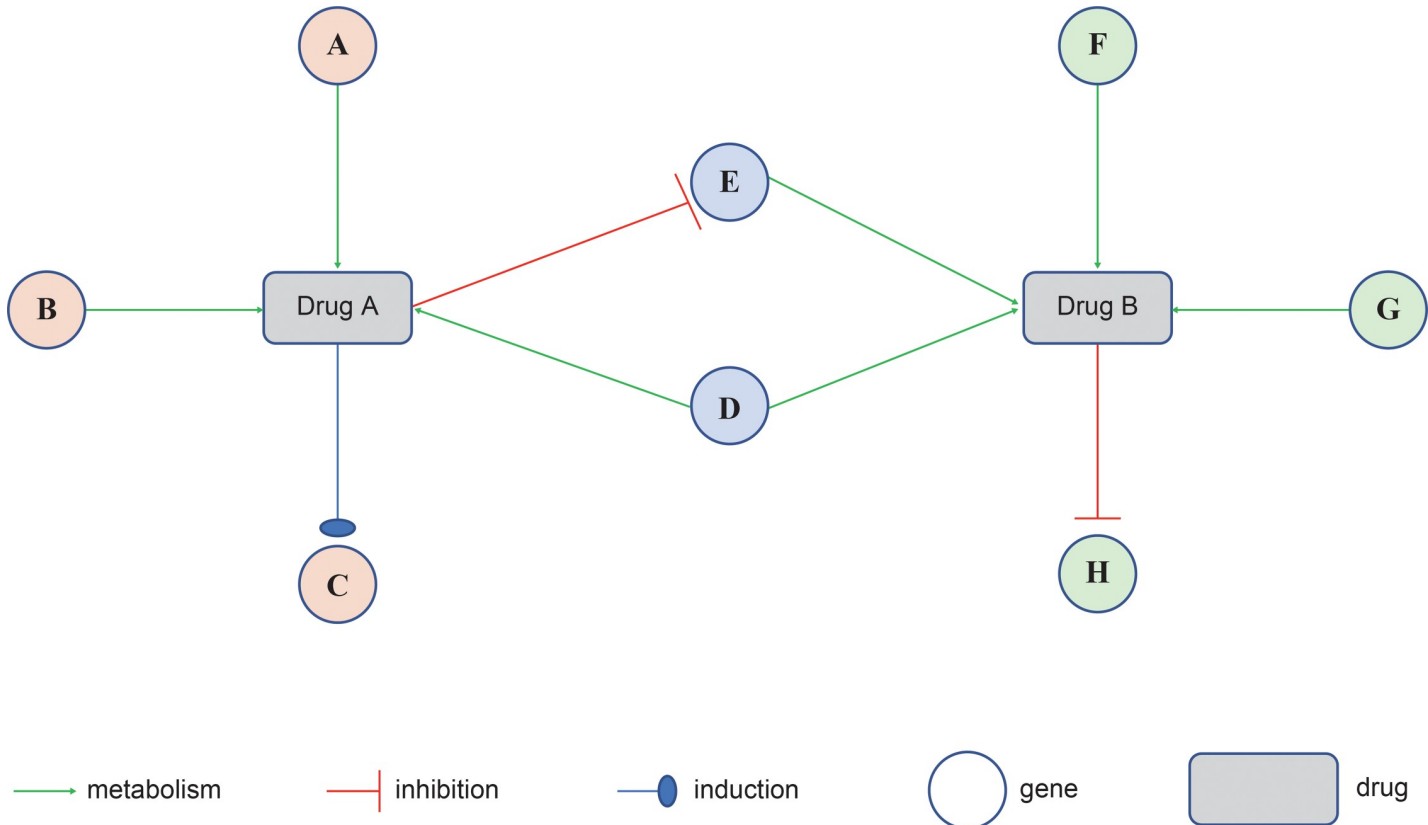

**Fig 4. Drug gene drug interaction between Drug A and Drug B.** If Drug A inhibits gene E and Drug B becomes the substrate of gene E, the metabolism of Drug B, which is the substrate, slows down when Drug A and Drug B are taken together.

However, this method is limited in the sense that we cannot assure the accuracy of the extracted gene set due to the absence of a gold standard; thus, we interpret a change in plasma drug concentration assuming that the gene set is correct. However, experimental or clinical studies can provide a more accurate way of approaching analyses compared to gold standards. [25]

## Future works

Drug reactions in the human body are not simple, and adverse drug reactions (ADRs) also frequently occur because individual differences in drug response are significant. To prevent ADRs, we attempted to design an approach to predict personal drug risk through pharmacogenomics. Because this approach addresses only genetic elements, it is difficult to reach a definite conclusion. Further studies require the creation of upgraded pathways by applying pharmacokinetic or clinical content to newly proposed drug pathways. This new approach to predicting drug risk poses many challenges. However, it is a step to go toward determining drug safety at the individual level.

## Supporting information

**S1 Table. Interaction types and action types on Clopidogrel from DrugBank 5.0.1.** (DOCX)

**S1 Fig. Method of merging a diagram with a background frame and image.**
(TIF)

**S2 Fig. Sub-windows in pathways.**
(TIF)

**S1 File.**
(DOCX)

## Acknowledgments

We thank LYI who illustrated the background images of the pathways and PCH who managed the pathway codes.

## Author Contributions

**Conceptualization:** Joo Young Hong, Ju Han Kim.

**Data curation:** Joo Young Hong.

**Methodology:** Joo Young Hong.

**Supervision:** Ju Han Kim.

**Visualization:** Joo Young Hong.

**Writing – original draft:** Joo Young Hong.

**Writing – review & editing:** Ju Han Kim.

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
