## [Decision Letter · Decision Letter 0]

23 Dec 2019

PONE-D-19-31446

PG-path: Modeling and Personalizing Pharmacogenomics-based Pathways

PLOS ONE

Dear Dr. KIM,

Thank you for submitting your manuscript to PLOS ONE. After careful consideration, we feel that it has merit but does not fully meet PLOS ONE’s publication criteria as it currently stands. Therefore, we invite you to submit a revised version of the manuscript that addresses the points raised during the review process.

We would appreciate receiving your revised manuscript by Jan 31 2020 11:59PM. To enhance the reproducibility of your results, we recommend that if applicable you deposit your laboratory protocols in protocols.io, where a protocol can be assigned its own identifier (DOI) such that it can be cited independently in the future. For instructions see: http://journals.plos.org/plosone/s/submission-guidelines#loc-laboratory-protocols

We look forward to receiving your revised manuscript.

Kind regards,

Giuseppe Novelli

Academic Editor

PLOS ONE

Journal Requirements:

1. Please note that PLOS ONE has specific guidelines on software sharing (http://journals.plos.org/plosone/s/materials-and-software-sharing#loc-sharing-software) for manuscripts whose main purpose is the description of a new software or software package. In this case, new software must conform to the Open Source Definition (https://opensource.org/docs/osd) and be deposited in an open software archive. Please see http://journals.plos.org/plosone/s/materials-and-software-sharing#loc-depositing-software for more information on depositing your software.

2. We note that Figure(s) [2] in your submission contain copyrighted images. All PLOS content is published under the Creative Commons Attribution License (CC BY 4.0), which means that the manuscript, images, and Supporting Information files will be freely available online, and any third party is permitted to access, download, copy, distribute, and use these materials in any way, even commercially, with proper attribution. For more information, see our copyright guidelines: http://journals.plos.org/plosone/s/licenses-and-copyright.

1.    You may seek permission from the original copyright holder of Figure(s) [2] to publish the content specifically under the CC BY 4.0 license.

3. Thank you for including your competing interests statement; "The authors have declared that no competing interests exist."

We note that one or more of the authors are employed by a commercial company:Cipherome. Inc

Reviewers' comments:

Reviewer's Responses to Questions

**Comments to the Author**

1. Is the manuscript technically sound, and do the data support the conclusions?

Reviewer #1: Yes

Reviewer #2: Yes

2. Has the statistical analysis been performed appropriately and rigorously? 

Reviewer #1: I Don't Know

Reviewer #2: N/A

3. Have the authors made all data underlying the findings in their manuscript fully available?

Reviewer #1: Yes

Reviewer #2: Yes

4. Is the manuscript presented in an intelligible fashion and written in standard English?

Reviewer #1: Yes

Reviewer #2: Yes

5. Review Comments to the Author

Reviewer #1: The present manuscript describes a novel pharmacogenomics-based pathway, called PG-path that could help users to assist personalized drug prescriptions and counseling for personalized drug therapy. There is a lot of statistics and informatics in this manuscript, and I'm not qualified to go into all the algorithms used to generate the software. However, I really enjoied reading the manuscript and do believe that instruments that visually show all the pharmacokinetic and pharmacodynamic processes could be of help in the field.

I have only some suggestions to improve the quality of the article:

1) It is not clear to this reviewer if PG-path will be a commercially available tool or a free downloadable one. Please clarify.

2) PG-path was designed based on available gene-drug information, and on data from the 1000 genome project. How can this pathway be continuosly updated with the growing body of information that NGS technology provides every day?

3) Variants of the gene sequence are not the only ones that can impair the function and the activity of proteins involved in PK and PD processes. Is there any way to implement the path with epigenetic modifications and gene expression levels?

Reviewer #2: The present manuscript entitled “PG-path: Modeling and Personalizing Pharmacogenomics-based Pathways” proposes a new method to improve, personalize and visualize pharmacogenomics pathways. Its approach is interesting and could enrich the pathways analysis tools. The manuscript is technically sound, written in clear and correct language, and the data support the conclusions.

However, this article could be improved in some domains.

1. It is not clear how users can access to these new personalized pathways, applying the method described in “Materials and Methods” paragraph. Is there a new tool, or is it being planned, to develop PK / PD pathways?

2. The authors used SIFT to predict the effects of genetic variants. Since we talk about predictions without any support coming from functional studies, having only one predictive program could be limiting. Other predictive programs should also be evaluated, for example, VarSome, an analysis tool for human genetic variation that reports prediction scores from 20 different algorithms.

3. The authors affirmed that this approach “addresses only genetic elements”. It would be advisable to speculate on this limit.

4. The use of “gene deleteriousness” is inappropriate. It is more correct to state that the genetic variants are deleterious.

6. PLOS authors have the option to publish the peer review history of their article (what does this mean?). If published, this will include your full peer review and any attached files.

Reviewer #1: No

Reviewer #2: No

---

## [Author Response · Author response to Decision Letter 0]

6 Feb 2020

Reviewer #1: 

1) It is not clear to this reviewer if PG-path will be a commercially available tool or a free downloadable one. Please clarify.

The objective of this study is modeling the pharmacogenomic pathway, so we do not determine yet whether PG-path will be available commercially or freely.

2) PG-path was designed based on available gene-drug information, and on data from the 1000 genome project. How can this pathway be continuously updated with the growing body of information that NGS technology provides every day?

We are currently working on the best methodology to incorporate the updates on time, though the detailed process will take time to determine.

3) Variants of the gene sequence are not the only ones that can impair the function and the activity of proteins involved in PK and PD processes. Is there any way to implement the path with epigenetic modifications and gene expression levels?

Epigenetic modifications are measured using chromatin immunoprecipitation (ChIP) and bisulfite-based methods, ChIP-chip and ChIP-seq, or RNA-seq. Gene expression is analyzed using microarray or RNA-seq. With our platform, the pathway is analyzed using DNA-seq. Therefore, it is not currently possible to apply epigenetic modification and gene expression level to PG-path. However, we will think carefully about these factors.

Reviewer #2: 

1) It is not clear how users can access these new personalized pathways, applying the method described in the “Materials and Methods” paragraph. Is there a new tool, or is it being planned, to develop PK / PD pathways?

This paper only covers the steps of proposing a method for producing individual pathways. If the number of pathways increases, we are considering the creation of a website that can show results on the screen by searching distinct pathways in HTML format.

2) The authors used SIFT to predict the effects of genetic variants. Since we talk about predictions without any support coming from functional studies, having only one predictive program could be limiting. Other predictive programs should also be evaluated, for example, VarSome, an analysis tool for human genetic variation that reports prediction scores from 20 different algorithms.

In this study, pathway personalization was performed using the GVB score, defined as the geometric mean of the SIFT scores for the set of coding variants in a gene. However, evaluating the GVB score itself by applying various predictive scoring algorithms to GVB is an enormous task and should be conducted separately from this study.

3) The authors affirmed that this approach “addresses only genetic elements”. It would be advisable to speculate on this limit.

As mentioned, changes in drug concentration are influenced not only by genetic factors but also by clinical and environmental factors. These various factors will be investigated through further research.

4) The use of “gene deleteriousness” is inappropriate. It is more correct to state that the genetic variants are deleterious.

We have revised the term as you advise.

---

## [Editor Report · Decision Letter 1]

13 Mar 2020

PG-path: Modeling and personalizing pharmacogenomics-based pathways

PONE-D-19-31446R1

Dear Dr. Kim,

We are pleased to inform you that your manuscript has been judged scientifically suitable for publication and will be formally accepted for publication once it complies with all outstanding technical requirements.

With kind regards,

Giuseppe Novelli

Academic Editor

PLOS ONE
---

## [Editor Report · Acceptance letter]

23 Mar 2020

PONE-D-19-31446R1 

PG-path: Modeling and personalizing pharmacogenomics-based pathways 

Dear Dr. Kim:

I am pleased to inform you that your manuscript has been deemed suitable for publication in PLOS ONE. Congratulations! Your manuscript is now with our production department. 

With kind regards,

on behalf of

Prof. Giuseppe Novelli 

Academic Editor

PLOS ONE